# Good Metabolic Control in Children with Type 1 Diabetes Mellitus: Does Glycated Hemoglobin Correlate with Interstitial Glucose Monitoring Using FreeStyle Libre?

**DOI:** 10.3390/jcm10214913

**Published:** 2021-10-24

**Authors:** Rocio Porcel-Chacón, Cristina Antúnez-Fernández, Maria Mora Loro, Ana-Belen Ariza-Jimenez, Leopoldo Tapia Ceballos, Jose Manuel Jimenez Hinojosa, Juan Pedro Lopez-Siguero, Isabel Leiva Gea

**Affiliations:** 1Pediatrics, Hospital Costa del Sol, 29603 Marbella, Spain; rocio.porcel.86@gmail.com; 2Endocrinology and Diabetes, Hospital de Algeciras, 11207 Cadiz, Spain; cristinaantunez1991@gmail.com; 3Pediatrics, Hospital Regional de Malaga, 29010 Malaga, Spain; maria.mora.sspa@juntadeandalucia.es; 4Pediatric Endocrinology, Hospital Universitario Reina Sofia, 14004 Cordoba, Spain; 5Pediatric Endocrinology, Hospital Regional Materno-Infantil de Malaga, 29011 Malaga, Spain; leotapiaceb@hotmail.com (L.T.C.); jmjhinojosa@hotmail.com (J.M.J.H.); lopez.siguero@gmail.com (J.P.L.-S.); isabeleiva@hotmail.com (I.L.G.)

**Keywords:** type 1 diabetes mellitus, pediatric diabetes, continuous glucose monitoring, time in 28 range, HbA1c, capillary blood glucose test

## Abstract

Background: Good metabolic control of Type 1 diabetes (T1D) leads to a reduction in complications. The only validated parameter for establishing the degree of control is glycated hemoglobin (HbA1c). We examined the relationship between HbA1c and a continuous glucose monitoring (CGM) system. Materials and methods: A cohort prospective study with 191 pediatric patients with T1D was conducted. Time in range (TIR), time below range (TBR), coefficient of variation (CV), number of capillary blood glucose tests, and HbA1c before sensor insertion and at one year of use were collected. Results: Patients were classified into five groups according to HbA1c at one year of using CGM. They performed fewer capillary blood glucose test at one year using CGM (−6 +/− 2, *p* < 0.0001). We found statistically significant differences in TIR between categories. Although groups with HbA1c < 6.5% and HbA1c 6.5–7% had the highest TIR (62.214 and 50.462%), their values were highly below optimal control according to CGM consensus. Groups with TBR < 5% were those with HbA1c between 6.5% and 8%. Conclusions: In our study, groups classified as well-controlled by guidelines were not consistent with good control according to the CGM consensus criteria. HbA1c should not be considered as the only parameter for metabolic control. CGM parameters allow individualized targets.

## 1. Introduction

Type 1 diabetes mellitus (T1D) is one of the most common chronic diseases in children. Due to its onset early in life and the lack of a definitive treatment, those affected live with the disease for a long time and, therefore, have a high burden of morbidity and mortality, since complications can arise both in the short and long term [1].

It has been shown that good metabolic control of the disease leads to a reduction in these complications [2,3]. The only currently validated parameter for establishing the degree of control of the disease is glycated hemoglobin (HbA1c), which, although providing very useful information, has a number of important limitations. It is an analytical parameter that reflects the average blood glucose values in the preceding two to three months. The main limitations are that it does not consider acute hypoglycemic and hyperglycemic events or the frequency and magnitude of intraday and interday blood glucose variability. Similarly, HbA1c values may be affected in situations such as anemia, hemoglobinopathies, or transfusions, among others [4,5].

In the 1980s, the Diabetes Control and Complications Trial (DCCT) demonstrated that those patients with T1D who were able to maintain HbA1c levels closer to those without diabetes had a lower incidence of microvascular and cardiovascular complications, both avoiding or delaying their onset (primary prevention) and slowing their progression (secondary prevention). In addition, it was found that initial metabolic control had a long-term influence on the subsequent clinical course, which was termed “metabolic memory”. These data have been validated 30 years after the initial study with the Diabetes Control and Complications Trial/Epidemiology of Diabetes Interventions and Complications Study at 30 years (DCCT-EDIC) [6].

Targets for good disease control based on HbA1c levels vary depending on the consensus used [1,7]. All these guidelines agree that targets should be individualized for each patient (Table 1).

Since the advent of continuous glucose monitoring (CGM) systems, we have gained more information on the variability of blood glucose and on acute hypoglycemic and hyperglycemic events, all while decreasing the number of capillary blood glucose tests required.

From the data obtained from the CGM system downloads, efforts were made to identify a set of data that would serve as criteria for good or poor control of the disease. Accordingly, the ATTD (Advanced Technologies and Treatments for Diabetes) congress, reached a new consensus in which 10 parameters were defined [4,8] (Table 2).

The FreeStyle Libre flash glucose monitoring system 1 (Abbott Diabetes Care, Witney, UK) is an established technology that measures interstitial fluid glucose levels. A sensor worn on the back of the upper arm takes a reading every minute that can be scanned using a hand-held reader or smartphone to receive a current glucose result along with historic results with a 15-min frequency. The FreeStyle Libre sensors are calibrated in the factory and have a wear time of up to 14 days without the need for the user to perform daily calibration using finger-prick tests [9].

The IMPACT and REPLACE studies [10,11] conducted with the FreeStyle Libre 1 sensor in adults with T1D or type 2 diabetes (T2D) on insulin therapy showed no decrease in HbA1c compared to SMBG. The SELFY study [12], performed in children and adolescents with T1D, reported a decrease in HbA1c from 7.9% to 7.4% after 8 weeks of sensor use compared to SMBG (*p* < 0.001). A meta-analysis [13] performed in 2019 analyzing results from 271 studies found a 0.55% (95% CI −0.70, −0.39) decrease in HbA1c 2–4 months after initiation of FreeStyle Libre 1 use in patients with T1D and T2D. In the 447 children and adolescents included, the mean decrease in HbA1c was 0.54% (95% CI −0.84, −0.23), and this improvement was maintained at 12 months. It was concluded that initiation of sensor use as part of diabetes management resulted in a decrease in HbA1c in adults and children with T1D and in adults with T2D.

The objective of our study is the evaluation of pediatric patients with T1D after one year of use of the Free Style Libre system, categorizing them by their HbA1c, in order to be able to know the differences in the number of capillary blood glucose controls before and after implantation, as well as CGM parameters that presents each category of HbA1c.

## 2. Material and Methods

### 2.1. Study Design and Participants

The funding of this study was in accordance with the regulations of the Official Gazette of the Andalusian Government (BOJA), resolution of 17 April 2018, regarding the organization of the Andalusian Health Service (SAS) to include CGM systems among the benefits provided by the Andalusian Public Health Service and a research project funded by the Andalusian Ministry of Health and Family (PIGE 0533-219).

This study was carried out from June of 2018 to September of 2019, following approval by the Ethics Committee of the Regional Hospital of Malaga. This prospective study was undertaken following insertion of the FreeStyle Libre 1 sensor in 191 pediatric patients with T1D. The inclusion criteria were presence of T1D with disease duration of more than one year, age between 4 and 18 years at the start of the study, and no previous experience using the FreeStyle Libre 1. Furthermore, only those patients who had more than 80% use of the sensor were included.

The exclusion criteria were having previously used interstitial blood glucose monitoring or having anemia or hemoglobinopathy that could constitute a bias.

Finally, we subdivided all patients in different groups according to cut-off points of HbA1c: Group 1 HbA1c ≤ 6.5%Group 2 HbA1c 6.5–7% (more than 6.5 and less than or equal to 7)Group 3 HbA1c 7–7.5% (more than 7 and less than or equal to 7.5)Group 4 HbA1c 7.5–8% (more than 7 and less than or equal to 8)Group 5 HbA1c ≥ 8%

### 2.2. Variables

The data were extracted using the LibreView^®^ platform (Abbott Diabetes Care, Witney, UK) one year after insertion of the FreeStyle Libre 1 sensor (Abbott Diabetes Care, Witney, UK), taking into account the last 14 days of use prior to the office visit. The parameters collected were those accepted in the consensus guidelines on the interpretation of CGM [4,5,7]: time in range (TIR), percentage of time below range (TBR), and coefficient of variation (CV), as well as average number of scans per day. In addition to these variables, we also analyzed sex, age, and the number of capillary blood glucose tests performed before sensor insertion and one year later. HbA1c values were also obtained before sensor insertion and at one year of use. The determination was made through a capillary blood sample using the DCA Vantage analyzer system (immunoassay technique) in the laboratory of the Regional Hospital of Malaga.

Capillary blood glucose measurements were collected after downloading the last 14 days of the glucometer in use prior to sensor insertion. Collection of capillary blood glucose readings at one year was performed by downloading the glucometer in use. For all patients, this could be done through the LibreView^®^ platform, as they were using the reader with a glucometer function with FreeStyle Optium^®^ capillary blood glucose strips.

Clinical data were collected through a written questionnaire completed by the primary caregiver and supervised by the healthcare team.

## 3. Statistical Analysis

All analyses were performed with R (R Core Team, 2020, University of Auckland, CAL, USA). Normality and homoscedasticity were tested using the Anderson–Darling and Fligner–Killeen tests, respectively. For quantitative variables, the statistics (mean and standard deviation) were reported, and for categorical variables, the absolute and relative frequencies were reported. To study the relationship between each of the quantitative variables of two groups or samples, the *p*-value associated with the Student’s *t*-test or nonparametric test such as Wilcoxon test, when normality was not proven were conducted. To study the relationship between each of the quantitative variables of several groups or samples, the p-value associated with the Kruskall–Wallis test was used. In the case of two categorical variables, Fisher’s test or the chi-square test was used.

## 4. Results

### 4.1. Difference in the Number of Capillary Blood Glucose Tests Performed per Day

The patients were classified into five groups according to their HbA1c at one year of using the FreeStyle Libre 1 sensor. The differences in the effect of the sensor were explored between the different groups and within the same group, before using the sensor and at one year of use (Table 3).

In most of the groups, the patients performed fewer capillary blood glucose tests one year after insertion of the sensor (mean: −6.0, standard deviation: 2.0). For statistical analysis, the Wilcoxon signed-rank test for paired samples was used. Statistically significant differences were found between the number of capillary blood glucose tests before and after the use of the sensor within each group (V = 23, *p*-value < 0.0001).

To assess the difference in the number of capillary blood glucose tests between the different groups, we used the Kruskal–Wallis nonparametric test. There were no significant differences between the groups (Kruskal–Wallis, chi-squared = 4.5977, standard deviation = 4.0, *p*-value 0.3311) (Figure 1).

### 4.2. Monitoring Parameters

TIR is defined as the time during which blood glucose levels are between two points, usually in the range 70–180 mg/dL or 70–140 mg/dL. Statistically significant differences were found in both TIRs 70–180 mg/dL and 70–140 mg/dL between the different groups (Table 3).

According to the aim of TIR ≥ 70%, 33.33% of group 1 achieved time in range, while 12% of group 2 and 9.37% of group 3 achieved time in range.

TBR is defined as the percentage of time in which blood glucose is ≤70 mg/dL. We found no statistically significant differences between the different HbA1c categories (Table 3).

CV is a measure of glucose variability derived from the standard deviation and the interquartile range. No statistical significance was observed between the different HbA1c categories (*p*-value 0.054) (Table 3).

Table 3 reports TBR, which allowed us to observe that the lower TIR was not due to the higher TBR, since in most categories it was close to the recommended one (less than 5%); thus, it is the highest time above range which would explain the shortest time in range that we observed in the data.

## 5. Discussion

According to the data extracted from the latest ISPAD consensus, good metabolic control based solely on HbA1c is considered to have a value less than 7% [1,7]. However, although group 1 (HbA1c ≤ 6.5%) had the highest TIR between 70 and 180 mg/dL, the mean of this percentage was 62.214%, below the value accepted by the CGM Consensus [4,8] for optimal control, which is set at TIR ≥ 70%. Group 2 (HbA1c 6.5–7%) (which would also be within the good control group per the ISPAD) had a mean TIR of 50.462%. Thus, we can conclude that the HbA1c accepted as optimal for good metabolic control (≤7%) does not correspond to an adequate TIR value (≥70%). The data obtained from our study indicate that the groups classified as well-controlled by the NICE (The National Institute for Health and Care Excellence, UK), ISPAD (International Society for Pediatric and Adolescent Diabetes, Berlin, Germany), and ADA (American Diabetes Association, Virginia, USA) guidelines (≤6.5%, ≤7%, and ≤7.5%, respectively) [1,7] are not consistent with good control according to the CGM consensus criteria. With these results, data are presented that support a new paradigm for metabolic control of T1D, in which the validity of HbA1c is not considered as the only parameter for metabolic control of the disease. The incorporation of CGM parameters allows for clear and individualized metabolic control targets in terms of hyperglycemia, hypoglycemia, and variability, underscoring the need for scalability of therapies that allow these targets to be achieved. Longer studies to correlate the recommended monitoring parameters with long-term macrovascular and microvascular complications are needed.

Several studies to date have attempted to correlate HbA1c with TIR in T1D patients. In 2019, two meta-analyses [14,15] were performed that sought evidence of this association. The first of these [14] studied 545 adult patients with T1D and compared HbA1c with different CGM parameters. It was concluded that CGM measures relevant to hyperglycemia (including TIR and mean glucose) were highly correlated with each other but moderately correlated with HbA1c, meaning that a particular TIR or change in a patient’s TIR could be associated with a wide range of HbA1c values. The second [15] included a total of 1137 adult patients with T1D in whom data correlating percentage of TIR and HbA1c were analyzed. It was concluded that there was a strong relationship between the two, with every 10% change in TIR resulting in a 0.8% change in HbA1c.

The percentage of TBR considered optimal is ≤5%. We found that the groups meeting this parameter were those with HbA1c 6.5–7%, 7–7.5%, and 7.5–8%, while the groups with HbA1c ≤ 6.5% and over 8% had higher values than those recommended.

We also noted a sharp decrease in the number of daily capillary glucose tests in all groups with the use of the FreeStyle Libre system 1, which is associated with greater convenience for the patient. Pediatric patients with T1D had a mean of 7–8 capillary blood glucose tests per day for metabolic control, with physical deterioration (skin of the hands) and the social stigmatization associated with the continuous handling of blood.

## 6. Conclusions

Harmonization of the recommendations for glycated hemoglobin and for time-in-range is lacking, as patients with glycated hemoglobin considered to be adequately controlled have lower time-in-range averages than those recommended for the pediatric population. Long-term studies correlating monitoring parameters with long-term complications are needed to identify monitoring targets that reduce macrovascular and microvascular complications.

## Figures and Tables

**Figure 1 jcm-10-04913-f001:**
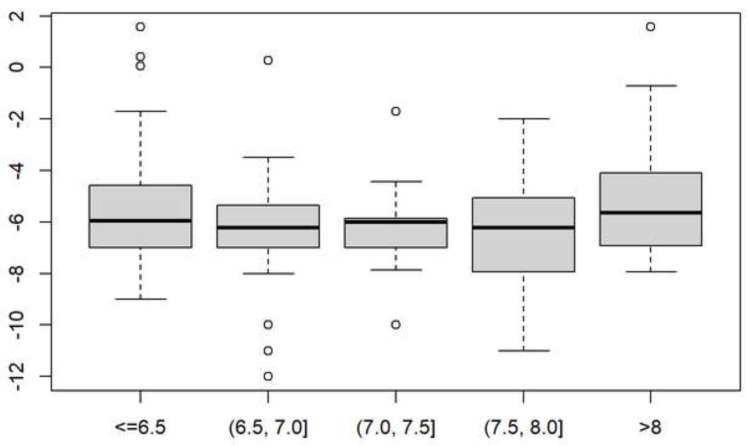
Difference in the lowest number of capillary blood glucose readings per day between the different capillary glycated hemoglobin groups one year after sensor insertion.

**Table 1 jcm-10-04913-t001:** Reference values for blood glucose and HbA1c according to different societies.

	NICE	ISPAD	ADA
Preprandial blood glucose (mg/dL)	70–126	70–130	90–130
Postprandial blood glucose (mg/dL)	90–162	90–180	
Bedtime glucose (mg/dL)	70–126	80–140	90–150
HbA1C (%)	≤6.5	<7	<7.5

NICE: The National Institute for Health and Care Excellence. ISPAD: International Society for Pediatric and Adolescent Diabetes. ADA: American Diabetes Association. HbA1c: glycated hemoglobin.

**Table 2 jcm-10-04913-t002:** Standardized CGM parameters in the ATTD 2019 consensus and target values for T1D.

Variable	Target
1. Number of days of CGM	14 days
2. Percentage of time CGM was active	>70%
3. Mean glucose/standard deviation	<154 mg/dL/<29%
4. Glucose management indicator/estimated HbA1C	<7%
5. Glucose variability (coefficient of variation) (%)	<36%
6. Time in range from 70 to 180 mg/dL (% of time)	>70%/>16 h 48 min
7. Time above range >180 mg/dL (% of time) Hyperglycemia level 1	<25%
8. Duration hyperglycemia level 1	<6 h
9. Time above range >250 mg/dL (% of time) Hyperglycemia level 2	<5%
10. Duration hyperglycemia level 2	<1h 12 min
11. Time below range <70 mg/dL (% of time) Hypoglycemia level 1	<4%
12. Duration hypoglycemia level 1	<1 h
13. Time below range <54 mg/dL (% of time) Hypoglycemia level 2	<1%
14. Duration hypoglycemia level 2	<15 min

CGM: continuous glucose monitoring. ATTD: Advanced Technologies and Treatments for Diabetes. HbA1c: glycated hemoglobin. T1D: type 1 diabetes.

**Table 3 jcm-10-04913-t003:** Patients categorized by level of glycated hemoglobin one year after sensor use with results of different variables.

		HbA1c ≤ 6.5% (*n* = 58)	HbA1c 6.5–7% (*n* = 49)	HbA1c 7–7.5% (*n* = 38)	HbA1C 7.5–8% (*n* = 28)	HbA1c ≥ 8% (*n* = 18)	*p*-Value (Kruskal–Wallis)
Sex	Boys	28 (48.3%) 30 (51.7%)	28 (57.1%) 21 (42.9%)	18 (47.4%) 20 (42.9%)	15 (53.6%) 13 (46.4%)	10 (55.6%) 8 (44.4%)	NS
	Girls						
Age (years) Mean (SD)		10.8 (3.3)	11.7 (3)	10.8 (3.4)	10.6 (2.5)	12.2 (2.3)	NS
No. capillary blood glucose/day baseline Mean (SD)		7.0 (1.4)	7.2 (1.4)	6.7 (1.1)	7.3 (1.6)	6.4 (1.1)	NS
Miss		9	5	9	1	2	
No. capillary blood glucose/day after a year Mean (SD)		1.3 (1.9)	0.9 (1.5)	0.7 (1.2)	1 (1.6)	1.4 (2.2)	NS
Miss		5	2	0	1	0	
% TIR (Glucose 70–180 ng/mL) Mean (SD)		62.214 (11.584)	50.462 (10.856)	47.625 (13.995)	39.385 (6.104)	32.636 (7.953)	<0.001
Miss		2	0	0	0	0	
% TIR (Glucose 70–140 ng/mL) Mean (SD)		40.923 (12.114)	30.885 (9.253)	26.781 (9.797)	22.538 (3.843)	17.636 (5.143)	<0.001
Miss		3	0	0	0	0	
% TBR Mean (SD)		5.397 (5.474)	4.271 (4.321)	3.789 (3.699)	3.667 (4.010)	5.167 (4.890)	NS
Miss		0	1	0	1	0	
CV Mean (SD)		38.562 (10.315)	38.983 (8.899)	40.170 (7.359)	37.544(6.224)	44.078 (8.917)	0.054
Miss		0	1	0	1	0	
Scanning frequency (SD)		9.857 (3.035)	9.120 (3.113)	10.875 (7.129)	11.538 (4.612)	7.545 (3.830)	0.238
Miss		2	1	0	0	0	

TIR: time in range measured in percentage. TBR: percentage of time below 70 mg/dL. CV: coefficient of variation measured in percentage. SD: standard deviation. NS: Not significant. HbA1c: glycated hemoglobin.

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
