# Peer review of "Good Metabolic Control in Children with Type 1 Diabetes Mellitus: Does Glycated Hemoglobin Correlate with Interstitial Glucose Monitoring Using FreeStyle Libre?"

_jcm, 2021, doi:10.3390/jcm10214913_

Round 1
Reviewer 1 Report
Thank you for the opportunity to review this manuscript examination the correlation between CGM glucose metrics versus HbA1c 1 year post sensor use in children and adolescents with type 1 diabetes. The authors included a moderately large sample of 191 patients, stratified by HbA1c categories and investigated relationships with time in range (TIR metrics).
I have a number of major concerns for the study, results and methods
i) The baseline characteristics of the patients pre CGM insertion (HbA1c, SMBG frequency etc should be provided).
ii) Table 1 The percentage sensor use, frequency of isCGM scanning at 1 year was not reported. This has ought to have a greater impact on HbA1c than frequency of SMBG? I am not entirely sure about the relevance of reduction of SMBG frequency during CGM on HbA1c.
iii) Critically, time above range was not reported (level 1 and level 2 hyperglycemia) in Table 1 or in the results or text. Is the lower TIR than expected due to greater TBR or TAR?
iv) Consider treating HbA1c and TIR as continuous variables for correlation analysis
v) Limitations of sensor accuracy and duration of CGM recording should be discussed. Could HbA1c also have been subject to bias due to anemia, haemoglobinapthies ?
Author Response
REVIEW TIR
Reviewer 1
Thank you for the opportunity to review this manuscript examination the correlation between CGM glucose metrics versus HbA1c 1 year post sensor use in children and adolescents with type 1 diabetes. The authors included a moderately large sample of 191 patients, stratified by HbA1c categories and investigated relationships with time in range (TIR metrics).
I have a number of major concerns for the study, results and methods
- The baseline characteristics of the patients pre CGM insertion (HbA1c, SMBG frequency etc should be provided).
We have another article pending of publishing in with we speak about the evolution of these items (basal, 3 months, 6 months and 1 year) . In this article the change of hbA1c wasn´t the aim of study, but it was used as one variable for classifying the studied subjects in 5 categories to correlate with parameters of CGM
According to SMBG frequency, it is reflected at table 3. (No. capillary blood glucose/day baseline and 1 year after)
- Table 1 The percentage sensor use, frequency of isCGM scanning at 1 year was not reported. This has ought to have a greater impact on HbA1c than frequency of SMBG? I am not entirely sure about the relevance of reduction of SMBG frequency during CGM on HbA1c.
We have added in inclusion criteria that only those patients who had more than 80% use of the sensor were included to avoid that the percentage of sensor use was considered a possible confounding factor in the interpretation of the data.
It has been added in table 3 the data of the number of scans that do not show significant differences between the different categories of glycosylated hemoglobin.
- Critically, time above range was not reported (level 1 and level 2 hyperglycemia) in Table 1 or in the results or text. Is the lower TIR than expected due to greater TBR or TAR?
We added the explanation in the article.
The time above the range has not been reported. Table 3 reports the TBR, which allows us to observe that the lower TIR is not due to the higher TBR, since in most categories it is close to the recommended one (less than 5%), so, it is the highest time above range which would explain the shortest time in range that we observe in the data.
- Consider treating HbA1c and TIR as continuous variables for correlation analysis
We have done it and there is a positive correlation as has been documented in other articles. But the objective that we wanted to share in this article is that glycosylated hemoglobin is a variable that overestimates adequate control and patients who are considered well controlled with glycosylated hemoglobin present time values ​​in range insufficient with respect to those proposed by the consensus.
- Limitations of sensor accuracy and duration of CGM recording should be discussed. Could HbA1c also have been subject to bias due to anemia, haemoglobinapthies ?
The MARD of Freestyle 1 could be a limitation for the study. It would be usefull to reproduce this study with FreeStyle Libre 2 which has less MARD and check the results.
We added that bias such as anemia or haemoglobinopathies are exclusion criteria in the study.
Reviewer 2 Report
The authors propose a compared analysis of HbA1c and different parameters calculated using common and accessible device for CGM, here FreeStyle Libre, in a pediatric population (4-18y) before and 1-year after they begin to use FSL.
The article is easy and nice to read, in a correct English.
The data are not very original, but still interesting, especially regarding a pediatric population. My greatest concern is the lack of specific question announced and addressed by the authors. At the end of a quite long introduction, the reader understands that he will hear about HbA1c and CGM parameters, but does not know what is/are the precise issues addressed. Then, the authors presents a series of relevant results, but they deserve a better red line.
Major comments
- Introduction: I strongly recommend to shorten the introduction and to add, at the end, what is (are) the precise question addressed by this paper, and the objectives
- Method 1: the article lacks a study flowchart (number of patients screened, attrition...). Do we must understand that all included patients performed the 1-year visit?
- Method 2 (statistical analysis): it's strange to read about Student and Mann-Whitney, and, the next page, to find a table with ANOVA or Kruskal-Wallis test (table 3) => please announce all the statistical tests in the Method and repeat it in the legend of the figure (if you keep it - see below)
- Table 3: the table needs improvement.
- p-values are not very relevant, as you compare not-randomized population which are intrinsically different regarding glucose homeostasis => with enough statistical power, one would except to see significant p-values at each line, so they are not informative. I would advise not to present these "p"
- in the discussion, you propose very quickly a threshold >70% for the TIR -> I advise to give this binary information in the table 3 and not only the mean of the TIR
- the numbers precision is too important, please simplify (number of daily capillary blood test : 7.0 and not 7.020, which is both clinically and statistically irrelevant here)
- TIR and TBR are first expressed as % or hours, and then you calculated a mean (SD)=> please specify the mean of what.
- HbA1c threshold: please express the borders correctly (< or ≤, so we can know where is a patient with HbA1C = 7%, for example)
- Discussion (lines 222-225) : "No significant differences were found between the greater number of capillary glycemia tests per day and the different glycated hemoglobin groups, which indicates that more frequent capillary glycemia testing is not associated with better metabolic control measured by glycated hemoglobin" : this affirmation is wrong. Your study presents a low statistical power (n = 191) which, additionally, you divided between 5 groups. So, if you fail to show a difference (p-value < 0.05 speaking), you just cannot conclude.
Minor comments
- General: Maybe global consistency could be improved following STROBE guidelines & checklist
- Introduction (line 42): the reference (ADA standards) is too general. Please quote the original article(s)
- Table 1 & 2: please give a proper legend, particularly for abbreviations (NICE, ISPAD, ADA, T1D..)
- Table 2: I had difficulties to read it, especially with the sign "<" added in different cells. I don't now if it's an editing issue but this must be corrected, for example using two cells when there are 2 informations (e.g <25% and < 6h)
- Method 1: please give the inclusion time (dates of first and last patients screened)
- Method 2 (lines 118-121): here, some exclusion criteria are a "mirror" of the inclusion criteria. Please remove redundant exclusion criteria
- Methods/Discussion: how did the authors choose to proposer 5 groups, and this specific thresholds for HbA1c (6.5/7/7.5/8%)? Maybe I missed it but this must appear somewhere in the method or discussion
- "qualitative variable": please use the more common "categorical variable"
- Figure 1 is very basic. If you use function hist() in R, I recommend at least to use argument breaks (=20 or 30, depending on display) or, for a best rendering, the ggplot2 package
Author Response
Reviewer 2
Comments and Suggestions for Authors
The authors propose a compared analysis of HbA1c and different parameters calculated using common and accessible device for CGM, here FreeStyle Libre, in a pediatric population (4-18y) before and 1-year after they begin to use FSL.
The article is easy and nice to read, in a correct English.
The data are not very original, but still interesting, especially regarding a pediatric population. My greatest concern is the lack of specific question announced and addressed by the authors. At the end of a quite long introduction, the reader understands that he will hear about HbA1c and CGM parameters, but does not know what is/are the precise issues addressed. Then, the authors presents a series of relevant results, but they deserve a better red line.
Major comments
- Introduction: I strongly recommend to shorten the introduction and to add, at the end, what is (are) the precise question addressed by this paper, and the objectives
Introduction has been shortened and the objectives has been added.
- Method 1: the article lacks a study flowchart (number of patients screened, attrition...). Do we must understand that all included patients performed the 1-year visit?
We added that we have not got any attrition along the study.
- Method 2 (statistical analysis): it's strange to read about Student and Mann-Whitney, and, the next page, to find a table with ANOVA or Kruskal-Wallis test (table 3) => please announce all the statistical tests in the Method and repeat it in the legend of the figure (if you keep it - see below)
We added it to Methods and table 3.
- Table 3: the table needs improvement.
We have tried to improve it.
- p-values are not very relevant, as you compare not-randomized population which are intrinsically different regarding glucose homeostasis => with enough statistical power, one would except to see significant p-values at each line, so they are not informative. I would advise not to present these "p"
We have removed no significtive p-values
- in the discussion, you propose very quickly a threshold >70% for the TIR -> I advise to give this binary information in the table 3 and not only the mean of the TIR
It is a very interesting suggestion. We have added this data in results.
- the numbers precision is too important, please simplify (number of daily capillary blood test : 7.0 and not 7.020, which is both clinically and statistically irrelevant here)
We have simplified daily capillary blood test.
- TIR and TBR are first expressed as % or hours, and then you calculated a mean (SD)=> please specify the mean of what. We have specified that it was %
- HbA1c threshold: please express the borders correctly (< or ≤, so we can know where is a patient with HbA1C = 7%, for example) We have changed it.
- Discussion (lines 222-225) : "No significant differences were found between the greater number of capillary glycemia tests per day and the different glycated hemoglobin groups, which indicates that more frequent capillary glycemia testing is not associated with better metabolic control measured by glycated hemoglobin" : this affirmation is wrong. Your study presents a low statistical power (n = 191) which, additionally, you divided between 5 groups. So, if you fail to show a difference (p-value < 0.05 speaking), you just cannot conclude.
The statement is withdrawn after consideration by the proofreader.
Minor comments
- General: Maybe global consistency could be improved following STROBE guidelines & checklist
STROBE Statement—Checklist of items that should be included in reports
|
|
Item No |
Recommendation |
|
Title and abstract |
1 |
(a) Indicate the study’s design with a commonly used term in the title or the abstract We provided study´s design in the abstract |
|
(b) Provide in the abstract an informative and balanced summary of what was done and what was found We provided what was done and found in abstract |
||
|
Introduction |
||
|
Background/rationale |
2 |
Explain the scientific background and rationale for the investigation being reported. We have explained scientific background |
|
Objectives |
3 |
State specific objectives, including any prespecified hypotheses We have stated objectives |
|
Methods |
||
|
Study design |
4 |
Present key elements of study design early in the paper We explainded study design in methods. |
|
Setting |
5 |
Describe the setting, locations, and relevant dates, including periods of recruitment, exposure, follow-up, and data collection We specified variable collection and follow-up time (1year) |
|
Participants |
6 |
(a) Give the eligibility criteria, and the sources and methods of selection of participants. Describe methods of follow-up We provided inclusion and exclusion criteria |
|
(b) For matched studies, give matching criteria and number of exposed and unexposed All patients were exposed |
||
|
Variables |
7 |
Clearly define all outcomes, exposures, predictors, potential confounders, and effect modifiers. Give diagnostic criteria, if applicable. We added as exclusión criteria potential confounders |
|
Data sources/ measurement |
8* |
For each variable of interest, give sources of data and details of methods of assessment (measurement). Describe comparability of assessment methods if there is more than one group All patients have the same type of freestyle and same variable of measures. |
|
Bias |
9 |
Describe any efforts to address potential sources of bias We added MARD as a possible bia |
|
Study size |
10 |
Explain how the study size was arrived at. The study size is composed by all the patient which was inserted Freestyle 1 in our department. |
|
Quantitative variables |
11 |
Explain how quantitative variables were handled in the analyses. If applicable, describe which groupings were chosen and why It was described in methods. |
|
Statistical methods |
12 |
(a) Describe all statistical methods, including those used to control for confounding It was described in methods. |
|
(b) Describe any methods used to examine subgroups and interactions It was described in methods. |
||
|
(c) Explain how missing data were addressed. We provided losses in each group. We do not get this data back. |
||
|
Results |
||
|
Participants |
13* |
(a) Report numbers of individuals at each stage of study—eg numbers potentially eligible, examined for eligibility, confirmed eligible, included in the study, completing follow-up, and analysed We report numbers of individuals at each group |
|
Descriptive data |
|
(a) Indicate number of participants with missing data for each variable of interest We indicated this. |
|
(b) Summarise follow-up time (eg, average and total amount) We indicated that our follow-up time is one year |
||
|
Outcome data |
15* |
Report numbers of outcome events or summary measures over time We provided summary measures over the time |
|
Main results |
16 |
(a) Give unadjusted estimates and, if applicable, confounder-adjusted estimates and their precision (eg, 95% confidence interval). Make clear which confounders were adjusted for and why they were included We have not included confounders. |
|
(b) Report category boundaries when continuous variables were categorized We reported category boundaries |
||
|
Other analyses |
17 |
Report other analyses done—eg analyses of subgroups and interactions, and sensitivity analyses We reported all analysis done |
|
Discussion |
||
|
Key results |
18 |
Summarise key results with reference to study objectives We summarized key results |
|
Limitations |
19 |
Discuss limitations of the study, taking into account sources of potential bias or imprecision. Discuss both direction and magnitude of any potential bias. We added MARD as an inescapable bia. |
|
Interpretation |
20 |
Give a cautious overall interpretation of results considering objectives, limitations, multiplicity of analyses, results from similar studies, and other relevant evidence. We interpreted results considering similarities and differences from other studies. |
|
Other information |
||
|
Funding |
22 |
Give the source of funding and the role of the funders for the present study and, if applicable, for the original study on which the present article is based We added funding information. |
- Introduction (line 42): the reference (ADA standards) is too general. Please quote the original article(s)
We have added reference to this line
- Table 1 & 2: please give a proper legend, particularly for abbreviations (NICE, ISPAD, ADA, T1D..)
We have added abbreviations.
- Table 2: I had difficulties to read it, especially with the sign "<" added in different cells. I don't now if it's an editing issue but this must be corrected, for example using two cells when there are 2 informations (e.g <25% and < 6h)
We divided information in several cells.
- Method 1: please give the inclusion time (dates of first and last patients screened)
- We have added it to methods.
- Method 2 (lines 118-121): here, some exclusion criteria are a "mirror" of the inclusion criteria. Please remove redundant exclusion criteria
We removed redundant exclusión criteria
- Methods/Discussion: how did the authors choose to proposer 5 groups, and this specific thresholds for HbA1c (6.5/7/7.5/8%)? Maybe I missed it but this must appear somewhere in the method or discussion
We have chosen those groups according to cut-off points proposed by the different scientific societies of good control.
- "qualitative variable": please use the more common "categorical variable"
We have changed it.
- Figure 1 is very basic. If you use function hist() in R, I recommend at least to use argument breaks (=20 or 30, depending on display) or, for a best rendering, the ggplot2 package
We would remove the figure, saying that it is redundant with the information included in the text